# Rice Husk as a Pore-Forming Agent: Impact of Particle Size on the Porosity and Diametral Tensile Strength of Porous Alumina Ceramics



T. T. Dele-Afolabi [1,2], M. A. Azmah Hanim [1,3,*], D. W. Jung [4,*], R. A. Ilyas [5], R. Calin [6] and A. R. Nurul Izzah [1]

1 Department of Mechanical and Manufacturing Engineering, Faculty of Engineering, Universiti Putra Malaysia, Serdang 43000, Malaysia
2 Department of Mechanical Engineering, Faculty of Engineering, Ajayi Crowther University, PMB 1066, Oyo 211213, Oyo State, Nigeria
3 Advance Engineering Materials and Composites Research Center (AEMC), Faculty of Engineering, Universiti Putra Malaysia, Serdang 43400, Malaysia
4 Department of Mechanical Engineering, Jeju National University, 1 Ara 1-dong, Jeju 690-756, Korea
5 School of Chemical and Energy Engineering, Faculty of Engineering, Universiti Teknologi Malaysia (UTM), Johor Bahru 81310, Malaysia
6 Department of Metallurgy and Materials Science Engineering, Faculty of Engineering, Kirikkale University, Kırıkkale 71450, Turkey
* Correspondence: azmah@upm.edu.my (M.A.A.H.); jungdw77@naver.com (D.W.J.)

**Abstract:** This study describes the porosity and particle size effects of rice husk pore former on the diametral tensile strength of porous alumina ($Al_2O_3$) ceramics. Porous $Al_2O_3$ ceramics with high porosity and sufficient diametral tensile strength were successfully prepared by the pore-forming agent method using rice husk (RH) as the pore former according to the sample formulation $Al_2O_3\text{-}^xRH_y$ (where 'x' denotes the particle size range in μm and 'y' denotes the percent weight content (wt%) of RH). The thermogravimetric analysis (TGA) and X-ray diffractometer (XRD) results revealed that silica was retained as rice husk ash in the developed porous $Al_2O_3$ after the decomposition of the starting rice husk pore former. Microstructures of the as-prepared porous $Al_2O_3$ ceramics having different RH additions exhibited hierarchical pore structures with increased particle size of the pore-forming agent. Porosity increased with larger particle size range of rice husk where the $Al_2O_3\text{-}^{63}RH_5$ demonstrated the least porosity (44.2 vol%), while the highest porosity (70.9 vol%) was demonstrated by the $Al_2O_3\text{-}^{125\text{--}250}RH_{20}$. The diametral tensile strength of the RH-shaped porous alumina ceramics declined from 16.97 to 0.65 MPa with increased particle size of the rice husk.

**Keywords:** porous materials; alumina; rice husk; microstructure; diametral tensile strength

## 1. Introduction

Porous ceramic systems with tailored pore structures have been widely used in industrial applications, including filtration of molten metal, wastewater treatment, biomedical implants, thermal insulation, and catalyst supports, owing to their outstanding thermal shock resistance, chemical stability, and mechanical properties [1–6]. Unlike porous materials produced from polymers that are hydrophobic, ceramics are highly hydrophilic. Hence, their preference over the polymer counterparts in the areas of application above. Consequently, developing viable techniques or new materials for the fabrication of robust and reliable porous ceramics has become a leading research trend over the past years. Meanwhile, most of the researches conducted thus far have emphasized the close connection between the fabrication method for infusion of pores into the ceramic matrix and the functionality of the resulting porous ceramic systems.

Among several preparation methods, the pore-forming agent (PFA) method has been described as a simple and flexible technique for producing porous ceramic materials

with hierarchical pore structures through the decomposition of the starting pore former within the ceramic matrix during the sintering phase [7–12]. Out of the various pore-forming agents available for preparing porous ceramic materials, organic materials have become a focal point lately [13–16]. Starches, for example, have attracted much attention because of their intrinsic capacity to demonstrate both the pore-forming and body-forming capabilities [17–20].

Porous $Al_2O_3$-20 wt% $ZrO_2$ ceramic composites shaped with corn starch demonstrated total porosity values and compressive strengths within the ranges of 9.5–65% and 290–60 MPa, respectively [21]. Prabhakaran et al. [22] employed wheat powder as a pore former and a gelling agent in producing porous $Al_2O_3$ with total porosity, pore dimension, and diametral tensile strength ranges of 67–76.7%, 200–800 μm, and 5.9–2.01 MPa, respectively. Talou et al. [23] achieved porosity and pore dimension values of 39–44% and 10–70 μm, respectively, in porous mullite ceramics shaped with pore formers from potato, corn, and cassava starches. More recently, studies [24–26], including our previous researches [25,26] reported the use of agricultural wastes, particularly corn cob and sugarcane bagasse in preparing porous $Al_2O_3$ ceramics with diverse ranges of porosity, pore structure, and tensile strength. Overall, an inverse relationship was reported between the mechanical properties and the porosity.

Given their relatively constrained pore size (≤180 μm) despite exhibiting high porosity, the use of porous ceramics prepared with starch materials or agricultural wastes poses serious concerns in application areas such as thermal insulation, solid oxide fuels, and regenerative medicine where open and hierarchical pore formations are essential for optimum performance. To address these concerns, this work is aimed at employing rice husk as the pore former in controlling the physical properties (porosity and pore size) of porous $Al_2O_3$ ceramics. Moreover, the synergistic utilization of diverse particle sizes and weight fractions of corn cob in preparing porous $Al_2O_3$ was reported in our previous investigation [25]. This investigation produced porous $Al_2O_3$ systems with controlled porosity and hierarchical pores.

Rice husk, the hard protective covering surrounding the paddy grain and a major by-product of rice milling decomposes at a significantly lesser temperature as compared with the high-temperature sintering of $Al_2O_3$. During decomposition, it produces a residue called rice husk ash (RHA) which is a potential source of amorphous reactive $SiO_2$ capable of reacting with $Al_2O_3$ to form mullite ($3Al_2O_3.2SiO_2$). More so, rice husk is cheap and can be efficiently refined into diverse particle sizes, which gives room for an economical means of controlling the pore size. Considering the great prospects demonstrated by rice husk and rice husk ash in their usage as pore formers in the currently limited literature [27–29], these materials are no doubt the most frequently utilized agricultural waste materials for preparing porous ceramics. In a previous study [30], different particle sizes of rice husk pore former were used to successfully prepare porous alumina ceramics. However, severe clustering of rice husk particles at higher weight fractions resulted in rapid deterioration of the mechanical properties of the porous alumina ceramics. More so, the diametral tensile strength, which is critical in determining the vulnerability of porous ceramics to tensile cracking, especially during post-fabrication processes, was not reported.

Therefore, for the purpose of gaining better insight into the prospects of the agricultural waste PFAs, the rice husk used in the current study was processed into different particle sizes to prepare porous alumina with hierarchical porosity. The microstructure of the developed porous $Al_2O_3$ was systematically characterized to elucidate the influence of porosity and different particle sizes of rice husk on the diametral tensile strength of the developed porous $Al_2O_3$ ceramics. Literature on the use of rice husk for porous ceramics development is still very limited and there is currently none that the authors are aware of that discusses the impact of different particle sizes of rice husk on the diametral tensile strength of porous alumina ceramics. Moreover, this study highlights a practical concept of inhibiting the clustering tendency of rice husk particles and improving their compatibility with ceramic matrix through treatment in acid solution.

## 2. Materials and Methods

### 2.1. Raw Materials

The raw materials were commercialized $Al_2O_3$ ($Al_2O_3$ > 99%, $D_{50} \leq 0.5$ μm; MHC Industrial Co. LFD, Xiamen, China) and rice husk waste (locally sourced from Nilai, Selangor, Malaysia). Sucrose solution was used as the binding agent since it is cheap and easily accessible.

### 2.2. Processing of Rice Husk Powder

To prepare the pore former in powdered form, rice husk was thoroughly washed to ensure removal of dust and later dried in an oven at 150 °C for 24 h to ease the pulverization process, which was performed using a laboratory blender (Waring 2-Speed Lab Blender, 22,000 rpm). To enhance the purity of the ash residue, rice husk powder was dried, treated in HCl solution at 70 °C for 2 h and sieved for particle classification by employing American Standard for Testing Materials (ASTM) test sieves to obtain particle size ranges of <63 μm, 63–125 μm and 125–250 μm, which are the most compatible for the preparation of the porous alumina ceramics according to a preliminary study. Thermal analysis was performed using the TGA/Differential scanning calorimetry (DSC) 1 HT analyzer (Mettler Toledo, Greifensee, Switzerland) at a rate of 5 °C/min (Table 1).

**Table 1.** TGA results of rice husk pore former.

| Feature | Result |
|:---:|:---:|
| Initial degradation temperature | 230 °C |
| 50% degradation temperature | 316 °C |
| Final degradation temperature | 345 °C |
| Ash content | 27.6% |

### 2.3. Sample Preparation

A general formula of $Al_2O_3$-$^xRH_y$ was utilized in expressing each sample composition, where 'x' denotes the particle size range in μm and 'y' denotes the percent weight content (wt%) of RH (i.e., 5, 10, 15, and 20 wt%). The raw materials were ball-milled for 60 min in a planetary mono mill (Fritsch Pulverisette 7) at 300 rpm in a container with 9 ceramic balls and mixed manually with sucrose solution. Thereafter, 4 g of the blend was uniaxially compacted in a steel die using a manual hydraulic press (Specac) at a pressure of 95 MPa to promote the production of crack-free green compacts. Lastly, the green compacts were exposed to a two-step heat treatment process; (i) pore former decomposition performed at a rate of 1 °C/min with gradual temperature increase (at each of 200, 300, 500, and 800 °C) and dwell of 1 h, and (ii) sintering at 1450 °C (5 °C/min). The schematic for developing the RH-shaped porous $Al_2O_3$ is presented in Figure 1.

### 2.4. Porosity Measurement

The density and porosity values of the samples were measured using the water immersion method based on Archimedes' principle (the weighing of dry and water infused samples in deionized water after boiling for 2 h and overnight cooling in water) with the following equations [31]:

$$\rho = \frac{m_{dry} \times \rho_{water}}{m_{wet} - m_{suspended} + m_{wire}} \tag{1}$$

$$P = \left(1 - \frac{\rho}{\rho_{theoretical}}\right) \times 100\% \tag{2}$$

where $\rho$ is the density, $m_{dry}$ is the dry mass of the sample, $m_{suspended}$ is the mass of the sample suspended in water, $m_{wet}$ is the mass of the sample after soaking $m_{wire}$ is the mass

of the suspending system, $\rho_{water}$ is the density of water, $\rho_{theoretical}$ is the theoretical density of the alumina, and $P$ is the volume fraction of the total porosity of the sample.

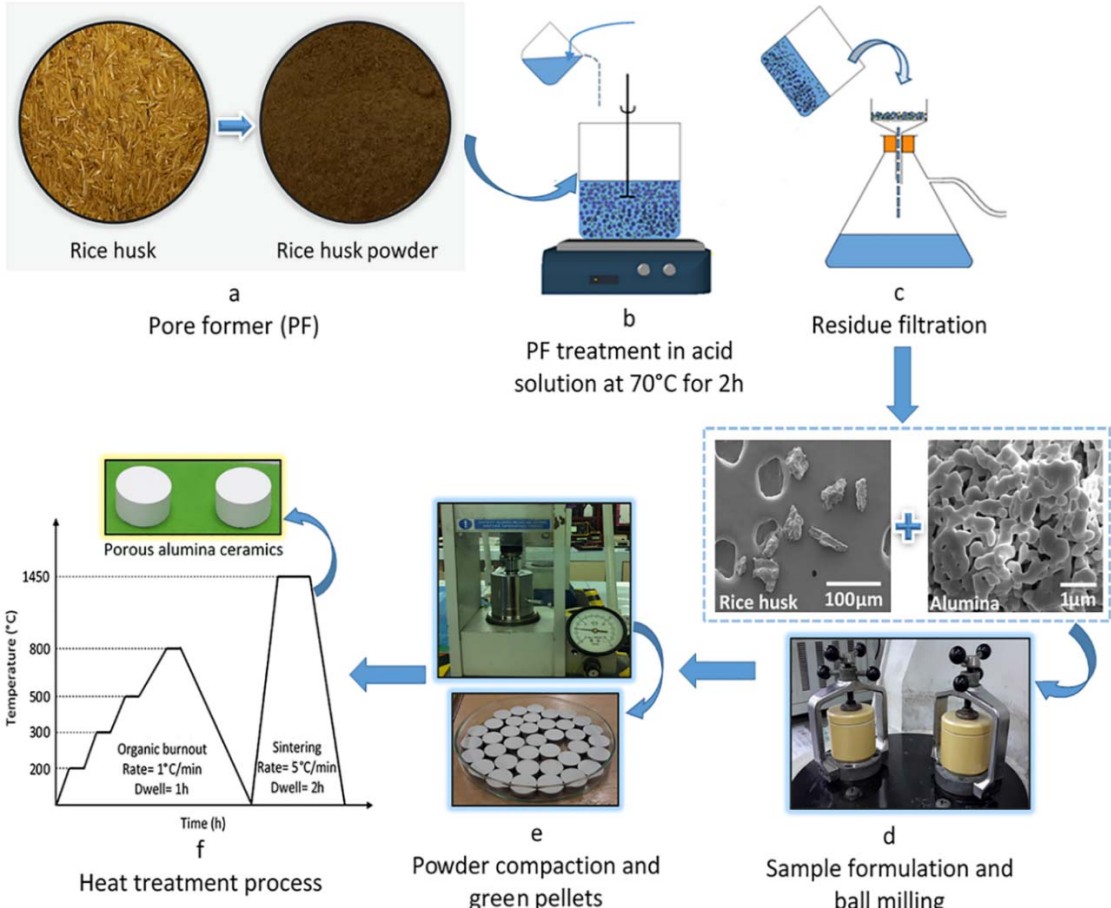

**Figure 1.** Schematic procedure for developing RH-shaped porous $Al_2O_3$ ceramics.

## 2.5. Microstructural Characterization

The microstructure and morphology of the pores were examined by field emission scanning electron microscope (FESEM) (Nova Nanosem 230, FEI Company, Hillsboro, OR, USA). By employing an image analysis software (ImageJ 1.46r, Java 1.6.0_20 64-bit), the pores were measured by averaging the sizes of more than 50 pores from approximately five FESEM images per sample formulation.

## 2.6. Elemental and Phase Characterization

The chemical composition of the RHA was characterized using the Wavelength Dispersive X-ray Fluorescence Spectrometer (WDXRF, S8 Tiger, Bruker, Billerica, MA, USA) and their quantitative chemical compositions were retrieved. The phase examination of the rice husk-shaped porous alumina ceramics was conducted using an X-ray diffractometer (Philips X'Pert Pro Panalytical, Almelo, The Netherlands), under CuK$\alpha$ radiation (wavelength = 1.5406 Å) at 40 kV and 40 mA. Thereafter, the retrieved data was analyzed using the PANalytical X'Pert Highscore Plus software (3.0.0.123).

## 2.7. Diametral Tensile Test

To ensure that the samples were tested under plane stress conditions, a sample dimension of 18.5 mm $\times$ 4.5 mm (diameter $\times$ thickness) was employed, which is in compliance with the geometry principle of $\frac{t}{D} \leq 0.25$ proposed in previous investigations [32,33]. A universal testing machine (INSTRON 3382, Illinois Tool Works, Glenview, IL, USA) operated at a speed rate of 0.1 mm/min as well as cardboard padding were utilized to perform

the test and ensure uniform stress distribution, respectively. The diametral tensile strength was estimated using Equation (3). At least five samples were tested, and the average value was reported.

$$\sigma = \frac{2F}{\pi D t} \tag{3}$$

where $F$ is the applied load; $t$ and $D$ are the thickness of sample and diameter of sample, respectively.

## 3. Results and Discussion

### 3.1. Phase Identification

The XRD pattern of the porous $Al_2O_3$ is shown in Figure 2. The primary phase in the porous ceramics is corundum ($Al_2O_3$), including a small amount of cristobalite ($SiO_2$). The emergence of the crystalline peak of $SiO_2$ in the porous alumina ceramics can be linked with the ash residue (rice husk ash; RHA) retained after the decomposition of the rice husk, where the weight content of the RHA was a quarter fraction (27.6%) of the starting RH pore former. In another study [30], $SiO_2$ from the retained RHA reacted with the alumina matrix to produce mullite during the heat treatment process at 1700 °C. In the present study, crystalline mullite peak is absent in the XRD pattern of the porous alumina.

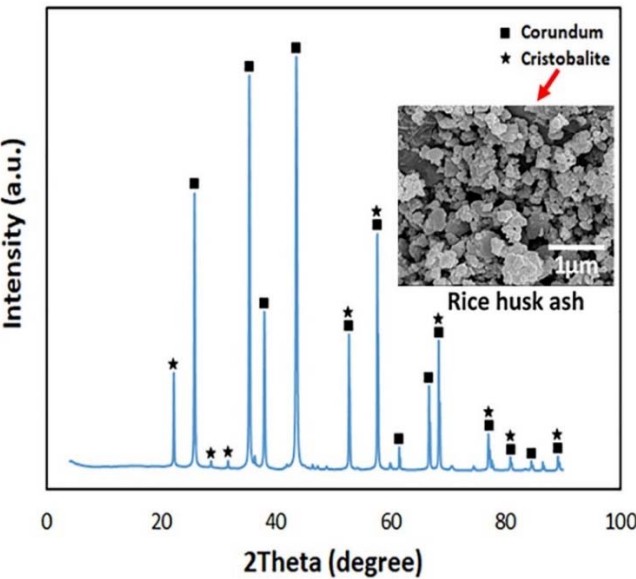

**Figure 2.** XRD pattern of RH-shaped porous $Al_2O_3$ ceramics.

The divergent findings in both studies can be rationalized as follows. On the one hand, the liquidus state of $SiO_2$ was exceeded at 1700 °C and the presence of a significant quantity of metastable liquid phase of $SiO_2$ behaved as a sink for dissolution of alumina, hence, resulting in the nucleation of mullite. On the other hand, the sintering temperature of 1450 °C utilized in the present study was not enough to melt the $SiO_2$, and neither was it suitable to hold the $SiO_2$ in an extremely reactive amorphous state. The explanation above can be rationalized further using the investigation of Johnson et al. [34], where the liquid $SiO_2/Al_2O_3$ interface was discovered as the locus for mullite formation. More so, controlled heat treatment below 900 °C has been recommended in the literature [35] for the preparation of highly reactive $SiO_2$ with superior adsorption capacity from rice husk waste. Despite the impossibility of achieving the mullite formation as earlier explained, the X-ray fluorescence results presented in Table 2 reveal that the acid treatment of rice husk effectively removed other metal oxides resulting in the enhanced purity of the $SiO_2$; 94.6 wt% and 99.7 wt% of $SiO_2$ for untreated RHA and treated RHA (see inset of Figure 2).

**Table 2.** Chemical composition of the rice husk ash.

| Materials | The Chemical Composition (wt%) | | | | | | | | | | | |
|---|---|---|---|---|---|---|---|---|---|---|---|---|
| - | SiO$_2$ | K$_2$O | P$_2$O$_5$ | CaO | SO$_3$ | MgO | Na$_2$O | Fe$_2$O$_3$ | Al$_2$O$_3$ | MnO | ZnO | Others |
| Untreated RHA | 94.6 | 1.76 | 0.86 | 0.79 | 0.73 | 0.58 | 0.45 | 0.09 | 0.08 | 0.08 | 0.02 | - |
| Treated RHA | 99.7 | 0.02 | 0.06 | 0.05 | - | - | - | 0.06 | 0.10 | - | - | 0.01 |

### 3.2. Porosity and Microstructure

The variations in the relative density and total porosity values with the weight content of the varying particle size ranges of rice husk are shown in Figure 3. As evidenced, the relative density and total porosity are inversely related. While the relative density of the porous samples developed with different particle sizes of RH decreased with increased RH addition from 5 to 20 wt%, the total porosity increased correspondingly. The reason for this has to do with the rapid increase in porosity gradient activated by the increased ratio of rice husk to alumina in the porous ceramics. In addition to this, the existence of intergranular porosity arising from partial sintering of porous alumina (at 1450 °C) was ascertained as an accompanying factor that aided the increase in porosity. In general, the Al$_2$O$_3$-$^{63}$RH$_5$ demonstrated the least porosity (44.2 vol%), while the maximum porosity (70.9 vol%) was recorded for the Al$_2$O$_3$-$^{125-250}$RH$_{20}$. It is also important to note that porosity increased with increasing particle size of rice husk for each RH loading. The reason for this is the relative reduction in the densification of alumina grains with the formation of wider pores in the porous alumina as the particle size range of the rice husk increases. This is consistent with findings elsewhere [30], where the densities of green porous ceramic compacts prepared using coarse rice husk particles were substantially lesser than those prepared using fine rice husk particles.

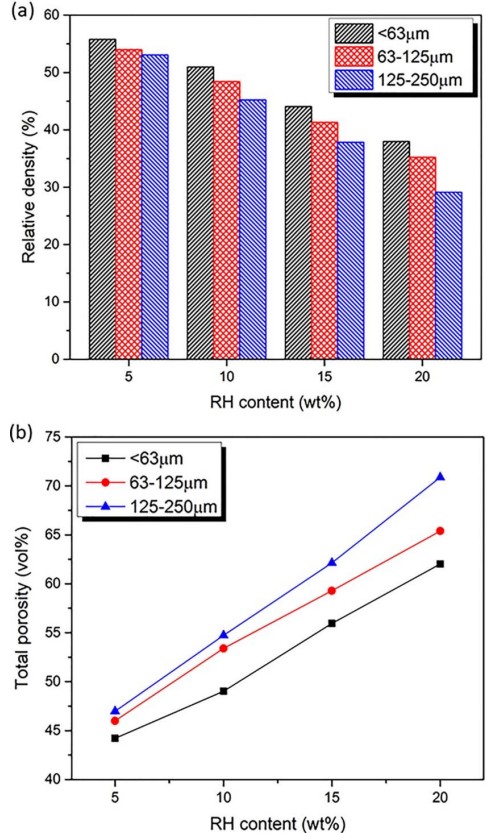

**Figure 3.** Variations of (**a**) relative density, and (**b**) total porosity of porous alumina ceramics with different weight contents of rice husk.

The microstructures of the porous $Al_2O_3$ ceramics developed with various sizes and additions of rice husk are shown in Figure 4a–f. The pores increased in size with increasing content of rice husk from 5 to 15 wt% for the different particle sizes. It is evident from the microstructures that the porous alumina ceramics prepared with 5 wt% RH exhibited isolated pore cavities relative to their counterparts prepared with 15 wt% RH. Moreover, the remarkable compactness of the pore walls, especially at higher additions of rice husk, can be ascribed to the retention of silica from rice husk ash (RHA) within the microstructure. Upon its retention, the RHA assisted in cushioning the adverse effect of severe dislocation of alumina grains on the densification of the pore walls.

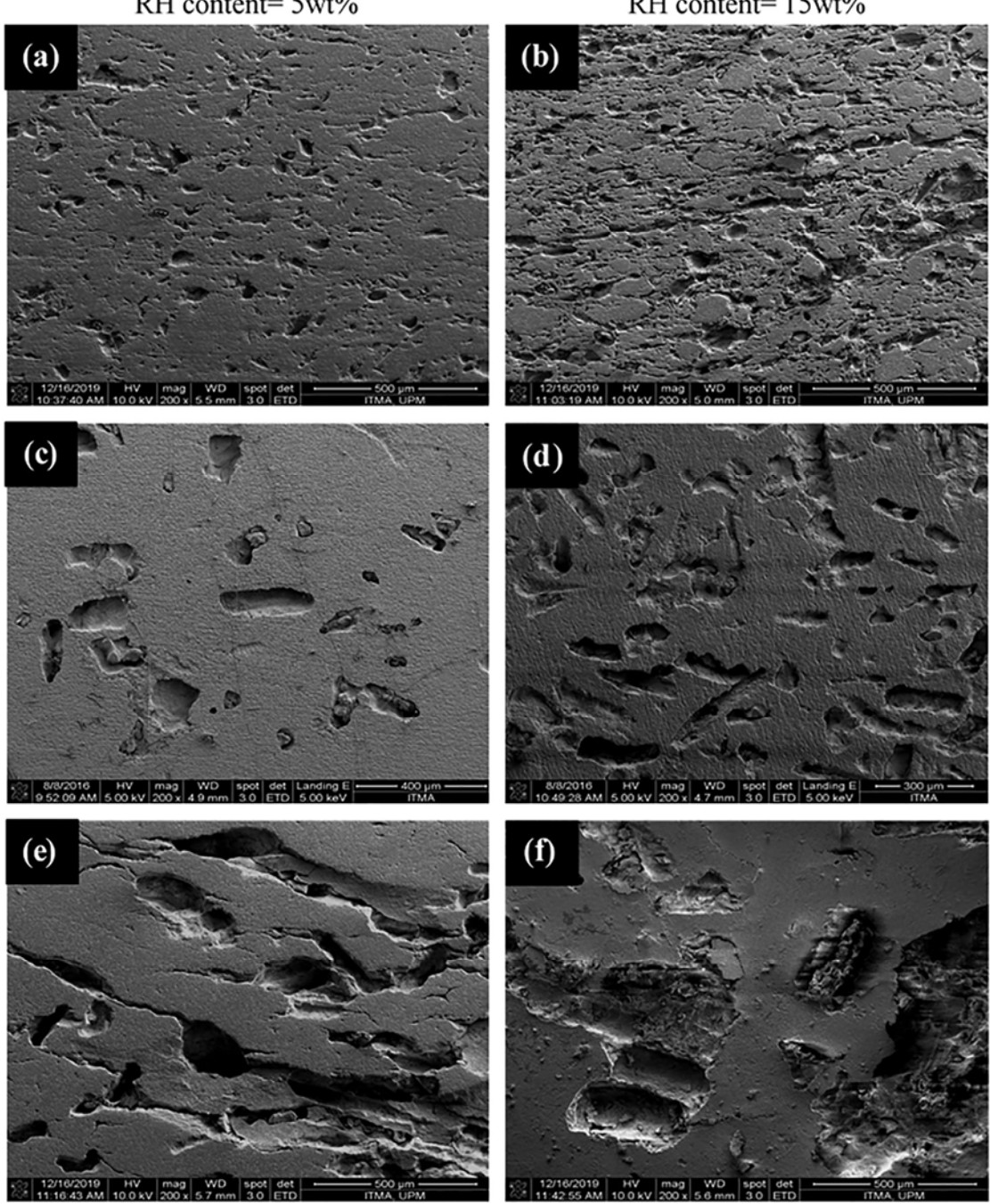

**Figure 4.** FESEM microstructures of porous $Al_2O_3$ prepared using (**a**,**b**) <63 μm, (**c**,**d**) 63–125 μm, and (**e**,**f**) 125–250 μm particle size ranges of rice husk.

As shown in Figure 5, the pores grew from 31.4–246.5 μm with increasing particle size range of RH from <63 μm to 125–250 μm. Furthermore, it is evident that the pore size of the porous $Al_2O_3$ increases with increasing weight fraction of RH. Evidently, from the microstructures, the samples developed with 5 wt% RH (i.e., $Al_2O_3$-$^{<63}RH_5$, $Al_2O_3$-$^{63–125}RH_5$, and $Al_2O_3$-$^{125–250}RH_5$) demonstrate isolated pores only, whereas the counterparts developed with 15 wt% have interconnected pores. Another interesting finding is that the mean sizes of the interconnected pores for the $Al_2O_3$-$^{125–250}RH_{15}$ and $Al_2O_3$-$^{125–250}RH_{20}$ are still within the 125–250 μm particle size range of the starting rice husk indicating the existence of a substantial quantity of the smaller particles within this range. Similar findings were reported by Prabhakaran et al. [22] and Zivcova-Vlckova et al. [36], where pore cavity size increased with increased additions of wheat starch and rice starch pore formers, respectively.

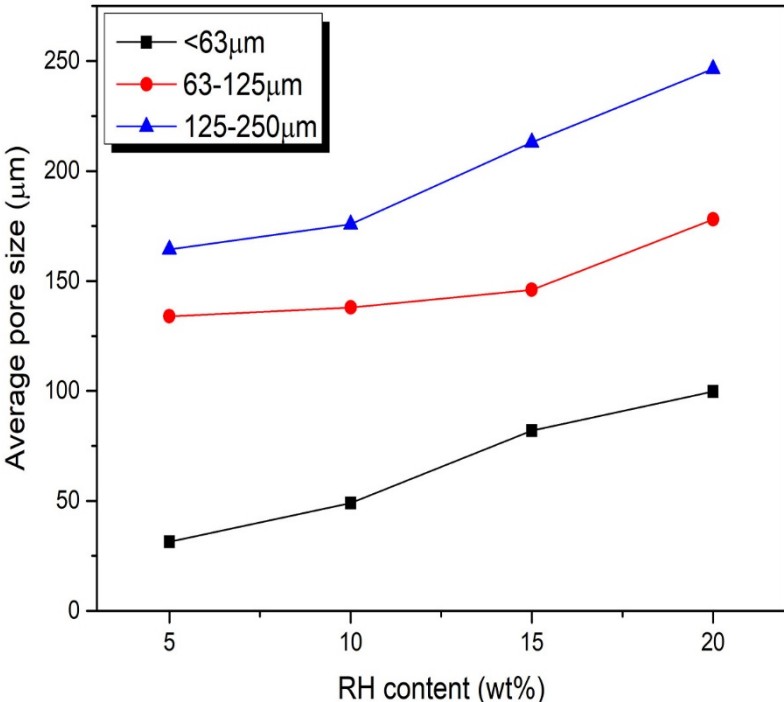

**Figure 5.** Variation in pore size of porous $Al_2O_3$ ceramics with varying weight fractions of rice husk.

Aside from the existential weak adhesion between fiber (i.e., rice husk) and matrix [37], the processing steps including: (i) the ball milling of $Al_2O_3$ and RH pore former, (ii) homogenization of the consequent blend with sucrose solution, and (iii) the powder compaction process induced the fragmentation of the larger particles in the 125–250 μm grade RH pore former. Hence, assisting the realization of the required optimum particle packing density for the preparation of crack-free sintered samples. The reason for the conclusion above contradicts those of other complementing studies [18,38], where the disintegration of starch pore former was suppressed by the starch consolidation casting processes of protein-aided foaming and granule swelling. Unfortunately, this technique is impracticable for the development of the RH-shaped porous $Al_2O_3$ due to the high immiscibility of rice husk with ceramic slurry.

### 3.3. Diametral Tensile Strength

After highlighting the physical properties of the RH-shaped porous $Al_2O_3$ samples above, a correlation between these properties and the diametral tensile strength is systematically substantiated here. Figure 6a and 6b show the diametral tensile strength of the porous $Al_2O_3$ as a function of RH addition and porosity. The diametral tensile strengths of the porous $Al_2O_3$ having porosity values between 44.2–70.9 vol% are within a range of

16.97–0.65 MPa. Overall, the $Al_2O_3$-$^{63}RH_5$ having a porosity of 44.2 vol% demonstrated the maximum strength of 16.97 MPa, whereas the least strength of 0.65 MPa was exhibited by $Al_2O_3$-$^{125–250}RH_{20}$ with the highest porosity of 70.9 vol%. Evidently, porosity increased correspondingly with increasing addition and particle size range of the rice husk, whereas the strength declined. This finding has been clearly substantiated in previous studies [39,40], where porosity and mechanical properties were inversely correlated.

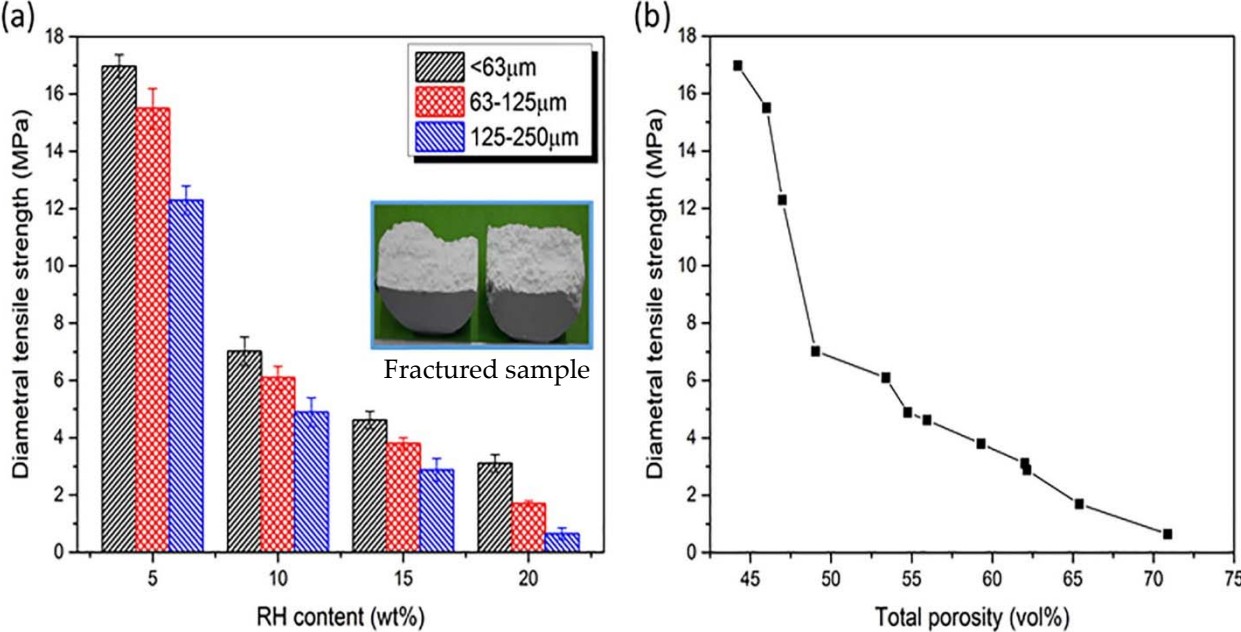

**Figure 6.** Diametral tensile strength of the porous $Al_2O_3$ ceramics as a function of (**a**) RH addition and (**b**) total porosity.

Irrespective of the RH particle size, it is observed that the diametral tensile strength decline intensified steadily between 5 and 20 wt% RH additions (see Figure 6a). Based on the primary features (porosity and pore size) employed in validating the mechanical strength of porous ceramic materials, the diametral tensile strength trend demonstrated by the RH-shaped porous $Al_2O_3$ ceramics can be closely related to the inherent influence of increased pore to matrix ratio; where with increased addition of rice husk, singular pores were substituted by open/meshed pores.

Even though the strength of the porous $Al_2O_3$ deteriorated as the pores grew bigger in samples developed with the same particle size range of rice husk, it is notable that the $Al_2O_3$-$^{125–250}RH_5$ sample displays comparatively superior strength and pore size of 12.3 MPa and 164 μm in comparison to the <63 μm and 63–125 μm categories having RH additions (up to 15 wt% and 20 wt%). It is, therefore, safe to say that porosity actuated by pores imbued after RH disintegration and intergranular porosity present in the skeletons owing to insufficient densification are the primary mechanisms influencing the mechanical reliability of porous materials. This, along with increasing pore interconnectivity, prominently substantiate the trend noticed in the finding above. In comparison to our previous study [25], where different particles of corn cob were employed as pore formers, the weaker glassy silicate component present in the RH-shaped porous alumina acted as a secondary nucleating site for micro-cracking. Hence, the relatively low diametral tensile strength values observed in the RH-shaped porous alumina ceramics developed in the current study as compared with their respective counterparts in the complementing study above.

With a view to ascertaining the effect of porosity and morphology of pores on the fracture mechanism inherent in the tested samples, two porous $Al_2O_3$ ceramics prepared with different weight percent of RH belonging to the same sample series were analyzed. Figure 7 shows the diametral tensile fracture morphologies of the $Al_2O_3$-$^{65–125}RH_5$ and

$Al_2O_3$-$^{65-125}RH_{15}$ porous $Al_2O_3$ ceramics. The fracture surface of the $Al_2O_3$-$^{65-125}RH_5$ appears to have a comparatively plain surface and a predominant cleavage fracture pattern (see marked region) that is marked by cracks propagating along the edges of the pores, as observed in Figure 7a. On account of this, it is safe to infer that the fracture mechanism in samples with isolated pores (i.e., 5–10 wt% RH loading) was controlled by the synergistic influence of pore structure and stress concentration at the silica protrusions (spot A, Figure 7c) within the $Al_2O_3$ matrix. However, for the $Al_2O_3$-$^{65-125}RH_{15}$, where porosity is way higher and pore interconnectivity intensified, fracture facets (see marked region) emanating from the interconnected pores can be observed on the fracture surface presented in Figure 7b.

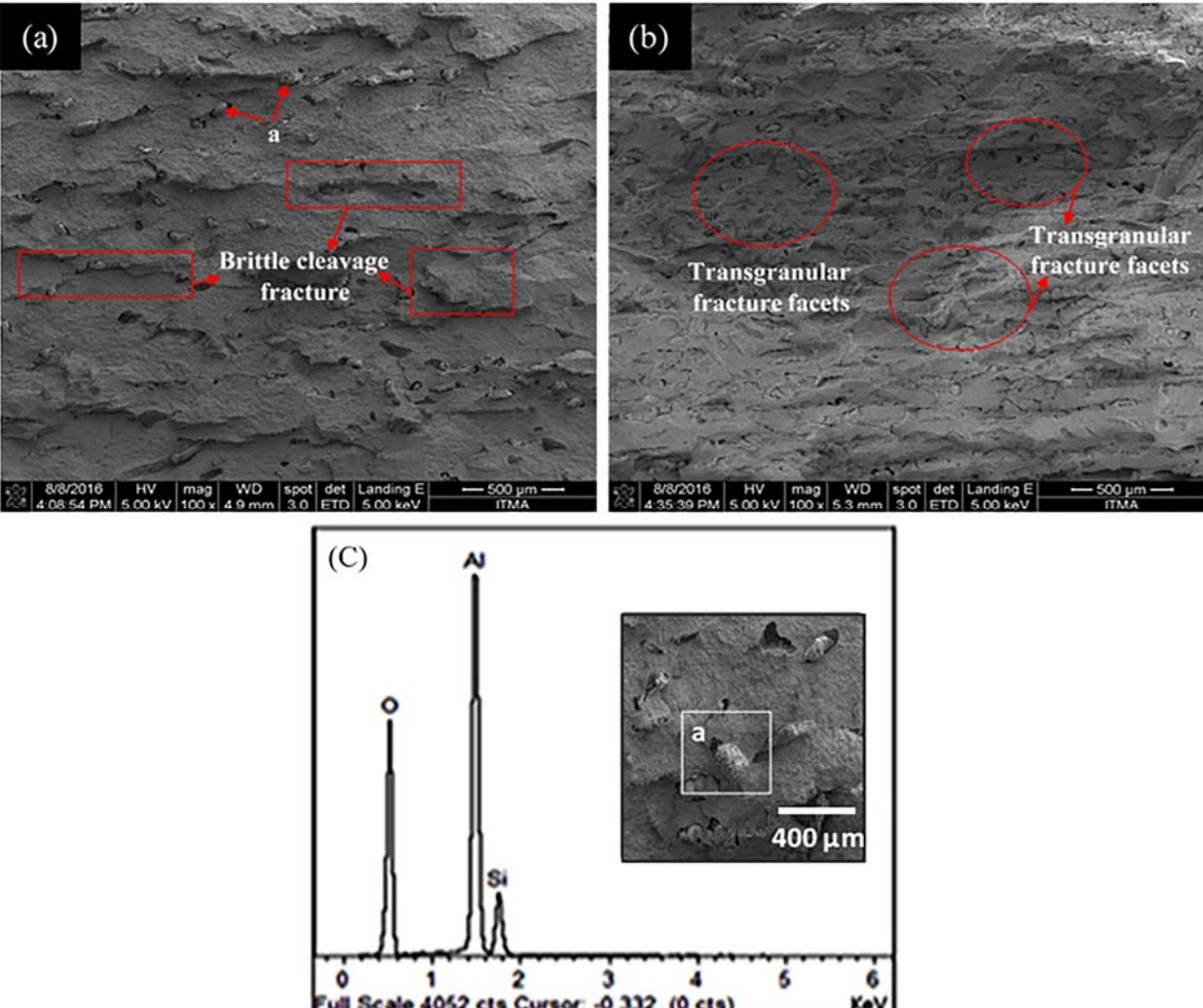

**Figure 7.** FESEM microstructures of fracture surfaces of (**a**) $Al_2O_3$-$^{63-125}RH_5$, (**b**) $Al_2O_3$-$^{63-125}RH_{15}$, and (**c**) EDS spectrum of spot A.

## 4. Conclusions

The present work has demonstrated the viability of developing porous $Al_2O_3$ ceramics with tailored porosity using rice husk as a pore former. The TGA and XRD results revealed that silica was retained as rice husk ash in the developed porous $Al_2O_3$ after the decomposition of the starting rice husk pore former. The total porosity and pore cavity size of the porous $Al_2O_3$ ceramics increased with increasing addition of RH pore former (5–20 wt%) from 44.2–70.9 vol% and 31.4–246.5 μm, respectively. Meanwhile, total porosity increased with larger particle size of RH pore former, with the $Al_2O_3$-$^{63}RH_5$ sample demonstrating the lowest total porosity of 44.2 vol% while the highest total porosity of 70.9 vol% was

demonstrated by $Al_2O_3$-$^{125–250}RH_{20}$. The diametral tensile strength of the porous alumina ceramics deteriorated with increasing porosity. The $Al_2O_3$-$^{63}RH_5$ sample demonstrated the highest tensile strength of 16.97 MPa while the $Al_2O_3$-$^{125–250}RH_{20}$ exhibited the lowest value of 0.65 MPa. Hence, new insights have been provided into the advanced utilization of rice husk waste for the preparation of porous alumina ceramics, which not only saves the cost of producing this porous ceramic system but also reduces agricultural waste pollution problems. The developed RH-shaped porous $Al_2O_3$ ceramics demonstrate remarkable potential as high-tech materials for thermal insulation, wastewater treatment, and solid oxide fuel cells.

**Author Contributions:** Conceptualization, T.T.D.-A. and M.A.A.H.; methodology, T.T.D.-A.; software, T.T.D.-A.; validation, T.T.D.-A., M.A.A.H. and R.A.I.; formal analysis, T.T.D.-A.; investigation, T.T.D.-A.; data curation, T.T.D.-A. and M.A.A.H.; writing—original draft preparation, T.T.D.-A.; writing—review and editing, T.T.D.-A., R.A.I., R.C. and A.R.N.I.; visualization, T.T.D.-A., R.A.I., R.C. and D.W.J.; supervision, M.A.A.H., R.C. and A.R.N.I.; funding acquisition, M.A.A.H. and D.W.J. All authors have read and agreed to the published version of the manuscript.

**Funding:** This research was funded by Universiti Putra Malaysia (UPM/GP-IPB/2020/9688700) and Jeju National University.

**Institutional Review Board Statement:** Not applicable.

**Informed Consent Statement:** Not applicable.

**Data Availability Statement:** The data presented in this study are available on request from the corresponding author.

**Acknowledgments:** This study and publication were supported by the Research Management Center, Universiti Putra Malaysia (UPM/GP-IPB/2020/9688700), and Jeju National University.

**Conflicts of Interest:** The authors declare no conflict of interest.

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
