# Peer review of "Rice Husk as a Pore-Forming Agent: Impact of Particle Size on the Porosity and Diametral Tensile Strength of Porous Alumina Ceramics"

_coatings, doi:10.3390/coatings12091259_

Round 1
Reviewer 1 Report
The current manuscript by Dele-Afolabi et al., has focused on the development of porous alumina ceramics while understanding the particle size effect of rice husk on porosity. The idea is not novel as there are already published reports on this topic. I have the following concerns which require addressed before this manuscript can be reconsidered.
MAJOR concerns:
Line 31-39: Authors have stated the use and importance of porous ceramics in various industrial applications. However, I think this section should explain some mechanistic insights as to why porous ceramics are better than their counterparts?
There are various reports where rice husk has already been exploited, I am appending below some papers; however, it is strongly advised that the authors should appreciate and cite the relevant literature and state the niche area within they want to explore? Why do they want to? And what will be their hypothesis?
1- Ali, M. S., Hanim, M. A., Tahir, S. M., Jaafar, C. N. A., Mazlan, N., & Amin Matori, K. (2017). The effect of commercial rice husk ash additives on the porosity, mechanical properties, and microstructure of alumina ceramics. Advances in Materials Science and Engineering, 2017.
2- Wei, Z., Li, S., Li, Y., Li, X., Xiang, R., & Xu, N. (2018). Porous alumina ceramics with enhanced mechanical and thermal insulation properties based on sol-treated rice husk. Ceramics International, 44(18), 22616-22621.
3- Liu, J., Ren, B. O., Lu, Y., Xi, X., Li, Y., Liu, K., ... & Huang, Y. (2019). Novel design of elongated mullite reinforced highly porous alumina ceramics using carbonized rice husk as pore-forming agent. Ceramics International, 45(11), 13964-13970.
4- Ali, M. S., Hanim, M. A., Tahir, S. M., Jaafar, C. N. A., Mazlan, N., & Amin Matori, K. (2017). The effect of commercial rice husk ash additives on the porosity, mechanical properties, and microstructure of alumina ceramics. Advances in Materials Science and Engineering, 2017.
Line 114: I am not satisfied by only giving reference to the ASTM method for porosity determination. Please explain and add a comprehensive method for its calculation? Did the authors determine the true density and solid fractions of the materials?
Line 116-117: Image analysis software? Which one and how this has been implemented and did the author validate the image analysis method?
Section 2.4: Please make sub-sections of characterisation methods and write separately with full details so these could be reproducible?
Section 4: I can understand the description of some main points from the results are important, but your section is totally missing what benefits one can have using the rice husk as pore former? So what if you made it? Please think critically and revise this section?
MINOR concerns
Line 89: Please add the blender name and conditions of grinding?
Line 91: ASTM? – full name, please
Line 92: Why this particle size range was selected?....Add reason
Line 93- Why DSC was not performed?
Line 103: add ball mill instrument name, and what conditions were used?
Line 104-105: Please describe the rationale for using sucrose solution and justification of compaction at 95 MPa?
Line 104-105: Please describe comprehensively the compaction process, conditions and instrument details?
Author Response
|
No |
Comment |
Answer |
|
1 |
Line 31-39: Authors have stated the use and importance of porous ceramics in various industrial applications. However, I think this section should explain some mechanistic insights as to why porous ceramics are better than their counterparts? |
Mechanistic insights have been added to the article on the priority placed on porous ceramics. |
|
2 |
There are various reports where rice husk has already been exploited, I am appending below some papers; however, it is strongly advised that the authors should appreciate and cite the relevant literature and state the niche area within they want to explore? Why do they want to? And what will be their hypothesis?
1- Ali, M. S., Hanim, M. A., Tahir, S. M., Jaafar, C. N. A., Mazlan, N., & Amin Matori, K. (2017). The effect of commercial rice husk ash additives on the porosity, mechanical properties, and microstructure of alumina ceramics. Advances in Materials Science and Engineering, 2017.
2- Wei, Z., Li, S., Li, Y., Li, X., Xiang, R., & Xu, N. (2018). Porous alumina ceramics with enhanced mechanical and thermal insulation properties based on sol-treated rice husk. Ceramics International, 44(18), 22616-22621.
3- Liu, J., Ren, B. O., Lu, Y., Xi, X., Li, Y., Liu, K., ... & Huang, Y. (2019). Novel design of elongated mullite reinforced highly porous alumina ceramics using carbonized rice husk as pore-forming agent. Ceramics International, 45(11), 13964-13970.
4- Ali, M. S., Hanim, M. A., Tahir, S. M., Jaafar, C. N. A., Mazlan, N., & Amin Matori, K. (2017). The effect of commercial rice husk ash additives on the porosity, mechanical properties, and microstructure of alumina ceramics. Advances in Materials Science and Engineering, 2017.
|
The suggested references have been added. However, we would like to state here that the novelty of our work lies in the utilization of acid treatment of the rice husk in mitigating the clustering tendency of this pore former at higher weight contents as well as in the influence of porosity and different particle sizes of rice husk on the diametral tensile strength of the developed porous Al2O3 ceramics. Also, these have been clearly stated in the last paragraph of the introduction. |
|
3 |
Line 114: I am not satisfied by only giving reference to the ASTM method for porosity determination. Please explain and add a comprehensive method for its calculation? Did the authors determine the true density and solid fractions of the materials? |
The method used for porosity measurement has been detailed and the density data has also been added to the article. |
|
4 |
Line 116-117: Image analysis software? Which one and how this has been implemented and did the author validate the image analysis method? |
The name of the software has been added as well as the implementation procedure. The validation was made with the few pores measured during the electron microscopy technique. |
|
5 |
Section 2.4: Please make sub-sections of characterization methods and write separately with full details so these could be reproducible? |
The suggestion has been effected. |
|
6 |
Section 4: I can understand the description of some main points from the results are important, but your section is totally missing what benefits one can have using the rice husk as pore former? So what if you made it? Please think critically and revise this section? |
The section has been reviewed as advised. |
|
7 |
Line 89: Please add the blender name and conditions of grinding?
|
The name and speed of the blender have been included. |
|
8 |
Line 91: ASTM? – full name, please
|
The full name has been added. |
|
9 |
Line 92: Why this particle size range was selected?....Add reason
|
The reason has been added. |
|
10 |
Line 93- Why DSC was not performed?
|
The TGA data gives the needed information regarding the degradation process of the rice husk which was used in establishing a suitable heat treatment profile for the preparation of the porous samples. Hence, our reason for not including the DSC data |
|
11 |
Line 103: add ball mill instrument name, and what conditions were used?
|
The information has been added. |
|
12 |
Line 104-105: Please describe the rationale for using sucrose solution and justification of compaction at 95 MPa?
|
The suggestion has been effected |
|
13 |
Line 104-105: Please describe comprehensively the compaction process, conditions and instrument details?
|
A comprehensive information has been given. |
Reviewer 2 Report
The article is well written and is suitable for publication in its current form.
Author Response
The article is well written and is suitable for publication in its current form.
Thank you Prof for your time and comment on the draft.
Reviewer 3 Report
The Article is devoted to the description of the method for preparing porous Al2O3 ceramics based on the use of rice husk as a pore former and the study of the crystalline and mechanical properties of this ceramic. It should be noted that porous ceramics are used in various fields and the search for promising materials is of undoubted practical interest. Research in this area has been carried out in various papers using corn cob, potato, corn and cassava starches and other agricultural wastes as pore formers. In particular, the porous ceramics based on rice husks were obtained and its properties were studied in [27]. Thus, the use of rice husks is not new and to justify the publication, the Authors should clearly explain the novelty of their work, and the advantages of their approach. As can be understood from the text of the article, the Authors used well-known techniques for the preparation and study of their samples. There is no comparative analysis of the results of using various pore formers.
After reading the article, it is clear, that the work contains incomprehensible pieces of text, stylistic errors. They are marked in yellow in the attached pdf file.
The Article may be considered for publication in Coatings if the Authors present the advantages of their approach for obtain the desired parameters for porous ceramic samples.

Author Response
|
No |
Comment |
Answer |
|
1 |
Thus, the use of rice husks is not new and to justify the publication, the Authors should clearly explain the novelty of their work, and the advantages of their approach. |
The novelty of our work lies in the utilization acid treatment of the rice husk in mitigating the clustering tendency of this pore former at higher weight contents as well as the in the influence of porosity and different particle sizes of rice husk on the diametral tensile strength of the developed porous Al2O3 ceramics. Also, this has been clearly stated in the last paragraph of the introduction. |
|
2 |
As can be understood from the text of the article, the Authors used well-known techniques for the preparation and study of their samples. There is no comparative analysis of the results of using various pore formers. |
Comparative analysis of the XRD results with another study has been made. Comparative analysis of the pore interconnection has been made with other studies were different pore formers were used. Similarly, another comparison has been made in section 3.3 with our previous study where different particle sizes of corn cob were used as pore formers. |
|
3 |
Line 24: What is the meaning of Al2O3-63RH5 |
The definition of the abbreviation has been added to the abstract. |
|
4 |
Line 43: What is the meaning of PFAs |
The meaning has been added. |
|
5 |
Line 69: Check sentence |
The paragraph has been reconstructed due to the comment raised by another reviewer. |
|
6 |
Line 145: What shows red arrow? Cristobalite? |
The arrow indicates the microstructure of the cristobalite (SiO2) which is present in the XRD pattern. |
|
7 |
Line 199: Why images have different scales. a,b,e,f, - 500 micrometers, but c, d - 400 and 300. |
All the microstructural images were taken with the same FESEM equipment using the same magnification of x 200. However, due to the long duration involved in conducting this research, the analysis was performed at different periods (over a year interval). Hence, the disparity in scale bar is likely to be as a result of recalibration of the FESEM equipment. |
|
8 |
Line 217-221: |
The sentence has been reconstructed. |
|
9 |
Line 222-226: |
The sentence has been reconstructed. |
Round 2
Reviewer 1 Report
The manuscript by Dele-Afolab et al. has focused on “Size effect of rice husk pore-forming agent on the porosity and diametric tensile strength of porous alumina ceramics”. This version a much improved. However, there are still some language issues. Please consider the following;
1. Please consider removing the language issues in the title. Maybe the following will give some help.
“Impact of rice husk particle size on the porosity and diametric tensile strength of porous alumina ceramics”
OR
Rice Husk as a Pore Forming Agent: Impact of Particle Size on the Porosity and Diametric Tensile Strength of Porous Alumina Ceramics”
2. There are still language issues (e.g. line 28- word rise is used for increased particle size, which is wrong it should be larger or increased) throughout the manuscript; please carefully proofread.
Author Response
|
No |
Comment |
Answer |
|
1 |
Please consider removing the language issues in the title. Maybe the following will give some help.
“Impact of rice husk particle size on the porosity and diametric tensile strength of porous alumina ceramics” OR Rice Husk as a Pore Forming Agent: Impact of Particle Size on the Porosity and Diametric Tensile Strength of Porous Alumina Ceramics”
|
The language issue in the title has been fixed according to the suggestions made |
|
2 |
There are still language issues (e.g. line 28- word rise is used for increased particle size, which is wrong it should be larger or increased) throughout the manuscript; please carefully proofread. |
The correction has been effected throughout the manuscript. |
Reviewer 3 Report
Unfortunately, according to the reviewer, the improved version of the Article does not meet the high requirements of the Journal. The Authors noted that the novelty of the approach is the use of acid to treat rice husks, but the benefits of this approach remain unclear. The pore sizes in the porous ceramics obtained by the Authors have a hierarchical structure, but this, as the Authors note, was also observed when using other types of pore formers. The conclusion that the use of rice provides new insights and demonstrates remarkable potential as high-tech materials for thermal insulation, wastewater treatment and solid oxide fuel cells can be found in previous works.
The corrected version of the Article contains unreadable pieces of text and stylistic errors.
According to the reviewer, the Article should not be published.
Author Response
|
No |
Comment |
Answer |
|
1 |
Unfortunately, according to the reviewer, the improved version of the Article does not meet the high requirements of the Journal. The Authors noted that the novelty of the approach is the use of acid to treat rice husks, but the benefits of this approach remain unclear. The pore sizes in the porous ceramics obtained by the Authors have a hierarchical structure, but this, as the Authors note, was also observed when using other types of pore formers. The conclusion that the use of rice provides new insights and demonstrates remarkable potential as high-tech materials for thermal insulation, wastewater treatment and solid oxide fuel cells can be found in previous works. |
This suggestion is well noted. However, in the last paragraph of the introduction which has been reconstructed, we have included the diametral tensile strength characterization as a focal point in our novelty statement.
To the best of our knowledge, our article is the first to investigate the effect of different particle sizes of rice husk pore former on the diametral tensile strength of porous alumina ceramics.
The advantage of our study lies in the acid treatment of rice husk to prevent severe clustering and the resultant rapid deterioration of mechanical strength as reported in other studies.
The suggestion that the hierarchical pore structures have been reported elsewhere using different types of pore formers is also well noted. However, the current study using rice husk with different particle sizes was performed in order to extend the purview of agricultural wastes as viable pore formers for preparing porous ceramics.
Hence, with the different ranges of rice husk used as pore formers, it is expected that there would be evolution of hierarchical pores in the developed porous ceramics which is the intent of this work ab initio. |
|
2 |
The corrected version of the Article contains unreadable pieces of text and stylistic errors.
|
The manuscript has been subjected to another round of proofreading |
Round 3
Reviewer 3 Report
Changes to the Article made by the authors and answers to the questions of the reviewer show that the article can be published.